# Glutamic acid–valine–citrulline linkers ensure stability and efficacy of antibody–drug conjugates in mice

Yasuaki Anami [1], Chisato M. Yamazaki [1], Wei Xiong[1], Xun Gui[1], Ningyan Zhang[1], Zhiqiang An[1] & Kyoji Tsuchikama [1]

Valine–citrulline linkers are commonly used as enzymatically cleavable linkers for antibody–drug conjugates. While stable in human plasma, these linkers are unstable in mouse plasma due to susceptibility to an extracellular carboxylesterase. This instability often triggers premature release of drugs in mouse circulation, presenting a molecular design challenge. Here, we report that an antibody–drug conjugate with glutamic acid–valine–citrulline linkers is responsive to enzymatic drug release but undergoes almost no premature cleavage in mice. We demonstrate that this construct exhibits greater treatment efficacy in mouse tumor models than does a valine–citrulline-based variant. Notably, our antibody–drug conjugate contains long spacers facilitating the protease access to the linker moiety, indicating that our linker assures high in vivo stability despite a high degree of exposure. This technology could add flexibility to antibody–drug conjugate design and help minimize failure rates in pre-clinical studies caused by linker instability.

[1] Texas Therapeutics Institute, The Brown Foundation Institute of Molecular Medicine, The University of Texas Health Science Center at Houston, 1881 East Road, Houston, TX 77054, USA. Correspondence and requests for materials should be addressed to K.T. (email: Kyoji.Tsuchikama@uth.tmc.edu)

Antibody–drug conjugates (ADCs) are an emerging class of chemotherapy agents with the potential to revolutionize current treatment strategies and regimens for cancers[1–4]. Indeed, the clinical success of ADCs has been demonstrated with FDA-approved ADCs for the treatment of patients with Hodgkin lymphoma (Adcetris®)[5, 6], HER2-positive breast cancer (Kadcyla®)[7, 8], acute lymphoblastic lymphoma (Besponsa®)[9], and acute myeloid lymphoma (Mylotarg®)[10] and more than 60 promising ADCs in clinical trials[11, 12]. The striking success has driven scientists and clinicians to further advance this molecular platform for developing effective therapeutics for cancers, microbial infection[13,] and immune modulation[14].

ADCs consist of potent drugs (payloads) linked to therapeutic monoclonal antibodies (mAbs) through chemical linkers. This molecular format enables pinpoint delivery of highly cytotoxic payloads to target tumor cells, resulting in greater potency, a broader therapeutic window, and more durable treatment effect than are possible with traditional chemotherapy agents alone[15, 16]. In addition to the choice of the antibody and payload, the ADC linker structure and antibody–payload conjugation modality impact ADC homogeneity, cytotoxic potency, tolerability, and pharmacokinetics (PK). These key parameters critically contribute to overall in vivo therapeutic efficacy[17–20]. Thus, refining linker and conjugation chemistries is of crucial importance for optimizing the therapeutic potential and safety profiles of ADCs.

Valine–citrulline (VCit) dipeptide linkers connecting a payload with a *p*-aminobenzyloxycarbonyl (PABC) group are standard cleavable linkers widely used in many successful ADCs including the FDA-approved ADC Adcetris[21, 22]. VCit linkers are cleaved by cathepsins upon internalization of ADCs by target cancer cells, resulting in traceless release of payloads (Fig. 1a)[17]. VCit linkers are stable in cynomolgus monkey and human plasma. However, it

has been reported that these linkers can be hydrolyzed in mouse plasma. The extracellular carboxylesterase 1c (Ces1c) has been reported as the enzyme responsible for this hydrolysis, which can result in premature release of toxic payloads in circulation prior to reaching tumors[23]. One approach to circumvent this problem is to use Ces1c-knockout mice (available from The Jackson Laboratory) or cross the Ces1c null alleles onto an immuno-compromised mouse model. However, the use of such genetically engineered mice may hamper smooth implementation of in vivo studies because of long lead time and limited choice of parent genetic background. The linker instability in mouse plasma can also be ameliorated by carefully selecting the linker attachment sites within an antibody and limiting the length of the VCit linker to minimize the exposure of the vulnerable moiety to extracellular enzymes, as demonstrated with several VCit-based ADCs[23–25]. However, it has also been confirmed that installation of VCit linkers at exposed conjugation sites and extending the linker structure result in rapid loss of payload in circulation[23, 25–28]. Almost all initial preclinical studies in drug development are performed using mouse models. Therefore, the instability creates an obstacle for evaluation of the therapeutic potential and safety profiles of VCit-based ADCs. In addition, this issue significantly limits flexibility in the choice of conjugation sites and linker design. Indeed, ADCs constructed using the multi-loading VCit linker recently developed by our group[29] were shown to have instability in mouse plasma and poor treatment efficacy in mouse breast tumor models (described in detail in the Results section).

Herein, we demonstrate that a glutamic acid–valine–citrulline (EVCit) tripeptide sequence provides exceptionally high long-term stability in mouse and human plasma while retaining the capacity to release the free payload upon cathepsin-mediated cleavage (Fig. 1a). We also demonstrate that an ADC constructed

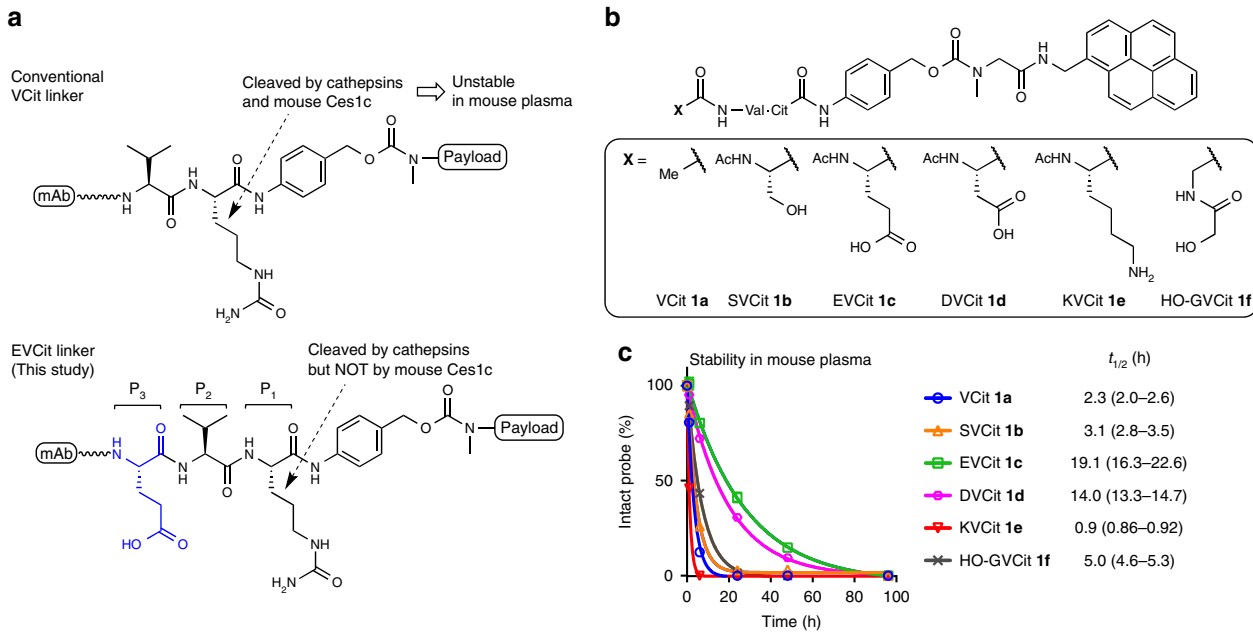

**Fig. 1** Structures and plasma stability of cathepsin-responsive cleavable peptides. **a** VCit and EVCit-based ADC linkers. VCit linkers are unstable in mouse plasma due to susceptibility to the extracellular carboxylesterase Ces1c. This instability often triggers premature release of payload in circulation. This study presents that VCit-based tripeptide sequences with an acidic side chain such as EVCit are responsive to cathepsin-mediated cleavage but highly stable in mouse plasma. **b** Structures of pyrene-based small-molecule probes containing a VCit (**1a**), SVCit (**1b**), EVCit (**1c**), DVCit (**1d**), KVCit (**1e**) or HO-GVCit (**1f**) cleavable sequence. **c** Stability of probes (**1a–f**) in undiluted BALB/c mouse plasma at 37 °C. (**1a**) blue circle; (**1b**) orange triangle; (**1c**) green square; (**1d**) magenta hexagon; (**1e**) red inversed triangle; (**1f**) gray cross. EVCit and DVCit probes (**1c**, **d**) showed great plasma stability while highly responsive to cathepsin B-mediated cleavage (see Supplementary Fig. 2). All assays were performed in triplicate. Error bars represent s.e.m. and values in parentheses are 95% confidential intervals

using the acidic EVCit linker exhibits by far greater in vivo stability and antitumor efficacy in xenograft mouse models bearing human breast cancer than does a VCit-based variant. Our findings indicate that the use of the EVCit linker system could minimize failure rates in preclinical studies using mouse models caused by linker instability, significantly expanding flexibility in designing ADCs. This technology may also provide a broadly applicable solution for enhancing stability and efficacy of other molecular classes of drug conjugates for targeted therapy.

## Results

**In vitro evaluation of cleavable tripeptide probes**. It has been reported that introducing a chemical modification to the N-terminus of the valine residue ($P_3$ position, Fig. 1a) significantly affects the plasma stability of VCit linker-based ADCs[23]. The authors demonstrated that VCit linkers with a hydrophilic group, such as 2-hydroxyacetamide group, at the $P_3$ position increased ADC stability in mouse plasma without impairing the reactivity to intracellular cathepsin B-mediated cleavage. Based on their findings, we speculated that installation of a highly polar functional group at the $P_3$ position could further enhance resistance to Ces1c-mediated degradation.

To test this hypothesis, we set out to assess a panel of peptide probes containing various amino acids at the $P_3$ position selected to confer plasma stability (Fig. 1b). We first synthesized VCit- and tripeptide probes **1a–f** by standard solid- and liquid-phase peptide synthesis and following carbamate formation (Supplementary Fig. 1). These model probes consisted of a XVCit–PABC unit where X is no amino acid (**1a**), serine (**1b**), glutamic acid (**1c**), aspartic acid (**1d**), lysine (**1e**), and hydroxyacetyl glycine (**1f**). Based on a report by Dorywalska and co-workers[23], we expected that probes **1b**, **f** would serve as mimics of the hydroxy-functionalized tripeptide ADC linker with increased mouse plasma stability. In particular, the 2-hydroxyacetamide group within probe **1f** is the modifier that provided the greatest stability in their report. Thus, we expected that comparing peptide sequences of interest with probe **1f** would clearly demonstrate the degree of improvement over the previous linker design. These linkers were covalently linked to 1-pyrenemethylamine, a surrogate of hydrophobic payloads with distinct UV absorbance at 342 nm[30]. Subsequently, probes **1a–f** were incubated in undiluted BALB/c plasma at 37 °C and the amount of each probe was monitored by HPLC (Fig. 1c). As anticipated, VCit probe **1a** exhibited a very short half-life ($t_{1/2} = 2.3$ h). SVCit probe **1b** was slightly more stable ($t_{1/2} = 3.1$ h) than VCit probe **1a**. However, the increase in stability was not as significant as what had been anticipated based on the previous report on the hydroxy-functionalized stable linker[23]. Intriguingly, acidic EVCit and DVCit probes **1c**, **d** showed greatly extended half-lives ($t_{1/2} = 19.1$ h and 14.0 h, respectively). The difference between these values is statistically significant as analyzed using GraphPad Prism 7, indicating that EVCit is superior to DVCit in terms of stability in mouse plasma. In contrast, basic KVCit probe **1e** showed the shortest half-life of all probes ($t_{1/2} = 0.9$ h). These results indicate that an acidic side chain at the $P_3$ position can effectively repel access of Ces1c, inhibiting the cleavage reaction. Probes **1a–e** tested were stable in human plasma and no significant degradation was observed after 2 days (Supplementary Fig. 2a). Given this point, the accelerated degradation seen for KVCit probe **1e** in mouse plasma may indicate that a basic side chain at the $P_3$ position provides an additional interaction with Ces1c, leading to fast bond cleavage. 2-Hydroxyacetamide probe **1f** was slightly more stable ($t_{1/2} = 5.0$ h) than SVCit probe **1b** but much less stable than EVCit and DVCit probes **1c**, **d**. Thus, the stabilizing effect of a neutral carbonyl group at the $P_3$ position was not as significant as that of a negatively charged carboxylic acid side chain.

Next, we tested probes **1a–e** for responsiveness to human cathepsin B-mediated cleavage (Supplementary Fig. 2b). SVCit probe **1b** showed a cleavage rate comparable with that of VCit probe **1a**. In contrast, we observed faster bond cleavage in EVCit, DVCit, and KVCit probes **1c–e** than in VCit and SVCit probes **1a**, **b**. These results demonstrated at the small-molecule level that the acidic EVCit and DVCit linker systems provided significantly enhanced stability in mouse plasma without impairing cathepsin B-mediated payload release.

**Construction and characterization of homogeneous ADCs**. To investigate whether the above-mentioned observation exists in the ADC format, we evaluated ADCs constructed using VCit-, SVCit-, and EVCit-based cleavable linkers. We first attached branched linker **2** (see Supplementary Fig. 3 for synthesis details) to an anti-HER2 mAb at glutamine 295 (Q295) by microbial transglutaminase (MTGase)-mediated conjugation according to the protocol developed by our group with minor modifications (Fig. 2a)[29]. The anti-HER2 mAb used for linker conjugation contained a mutation of the asparagine 297 into alanine (N297A)[31]. This mutation allows omission of removal of the N-glycan chain on N297, a step required for MTGase-mediated antibody–linker conjugation[32]. The conjugation afforded a homogeneous mAb–branched linker conjugate in high yield (Fig. 2b). In parallel, we synthesized SVCit- and EVCit-based modules containing monomethyl auristatin F (MMAF), polyethylene glycol (PEG) spacer, and dibenzocyclooctyne (DBCO) as a reaction handle for following strain-promoted azide–alkyne click reaction (Supplementary Fig. 4). A DBCO–VCit module was obtained from a commercial source. The number of PEG units in each module was adjusted so that all payload modules had a similar length ($PEG_4$ for the dipeptide VCit and $PEG_3$ for the tripeptides SVCit and EVCit). Each clickable module could be quantitatively conjugated to the common mAb–branched linker conjugate to give highly homogeneous ADCs with an average drug-to-antibody ratio (DAR) of 3.9 (determined by reverse-phase HPLC, Supplementary Fig. 5). We also prepared non-cleavable branched ADC **4** (DAR: 3.9) and an isotype control constructed using the EVCit–MMAF module (**5**, DAR: 3.9) in the same manner (Supplementary Fig. 6). Size-exclusion chromatography (SEC) analysis revealed that all ADCs produced existed predominantly in the monomer form (Fig. 2c). We also evaluated long-term stability by incubating each ADC at 37 °C in PBS (pH 7.4) for 28 days. No significant degradation or aggregation was observed by SEC analysis (Supplementary Fig. 7).

To assess how a chemical modification at the $P_3$ modification influences ADC hydrophobicity, we performed hydrophobic interaction chromatography (HIC) analysis under physiological conditions (phosphate buffer at pH 7.4, Fig. 2d). Highly polar EVCit ADC **3c** was detected earlier in retention time than VCit ADC **3a**, whereas the hydroxy group within SVCit ADC **3b** marginally affected the ADC hydrophobicity (Fig. 2e). These results demonstrate that constructing ADCs using carboxy-functionalized EVCit linkers can reduce ADC hydrophobicity at physiological pH. This feature is advantageous for the construction of ADCs, especially high-DAR ADCs, because high ADC hydrophobicity is known to trigger aggregation and fast clearance from the body[33].

**Validation of ADCs in vitro**. To investigate how ADC in vitro properties are modulated by a chemical modification to the $P_3$ position, we first evaluated ADCs **3a–c** for cathepsin B-mediated cleavage. Each ADC was incubated in the presence of human liver cathepsin B at 37 °C. The half-lives of VCit ADC **3a**, SVCit ADC **3b**, and EVCit ADC **3c** were determined to be 4.6 h, 5.4 h, and

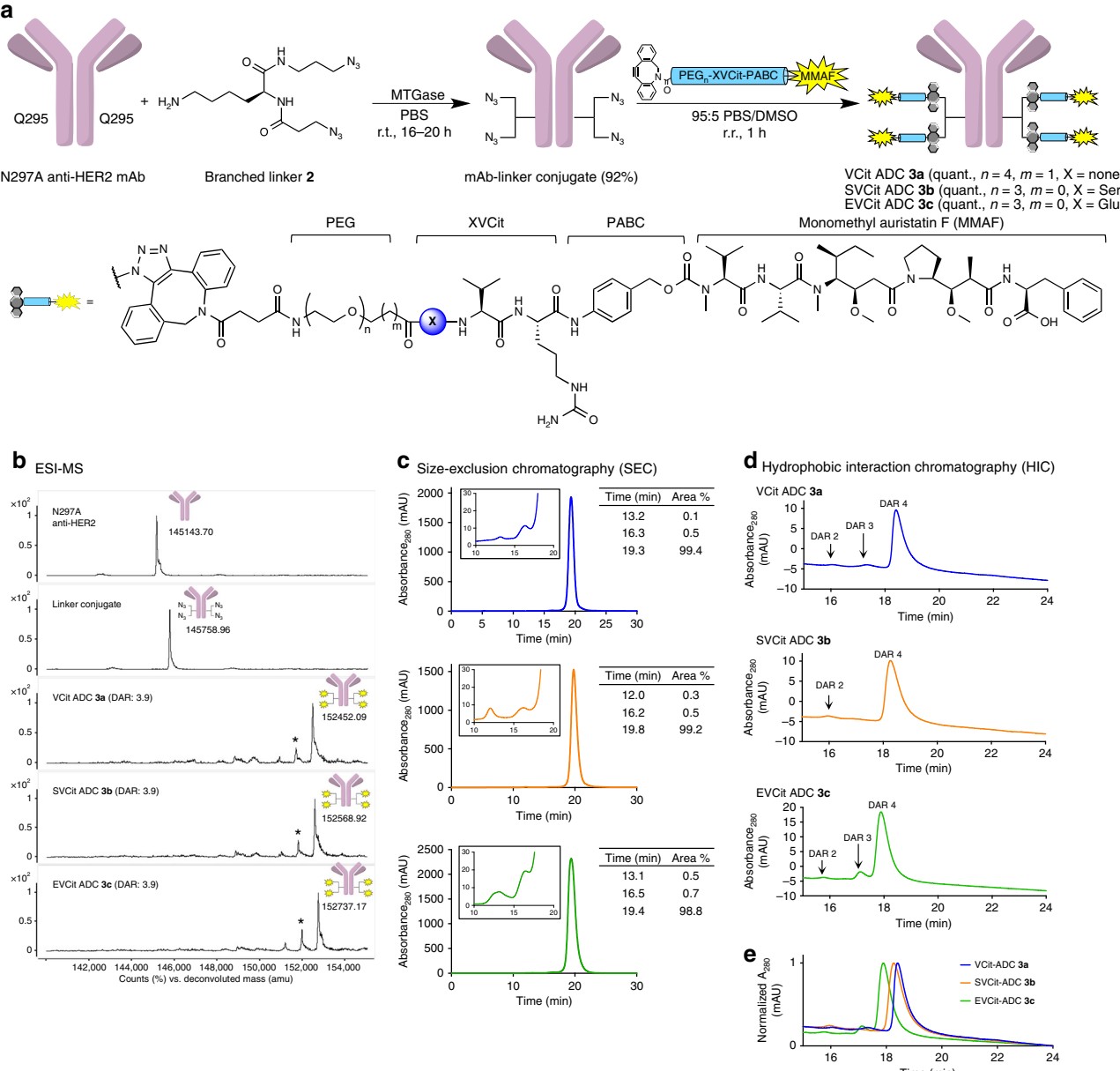

**Fig. 2** Construction and characterization of ADCs **3a**–**c**. **a** Construction of ADCs (**3a**–**c**) by MTGase-mediated branched linker conjugation and following strain-promoted azide–alkyne cycloaddition (cyan cylinder: PEG spacer–XVCit–PABC module; yellow spark: MMAF). **b** Deconvoluted ESI-mass spectra. Top panel: N297A anti-HER2 mAb. Second panel: antibody–branched linker conjugate. Third–fifth panels: highly homogeneous ADCs (**3a**–**c**). Asterisk (*) indicates a fragment ion detected in ESI-MS analysis. **c** SEC traces of ADCs (**3a**–**c**). **d** HIC analysis of ADCs (**3a**–**c**) under physiological conditions (phosphate buffer, pH 7.4). **e** Overlay of the three HIC traces (VCit ADC **3a**: blue; SVCit ADC **3b**: orange; EVCit ADC **3c**: green). DAR, drug-to-antibody ratio; MTGase, microbial transglutaminase; PABC, *p*-aminobenzyloxycarbonyl; PEG, polyethylene glycol

2.8 h, respectively (Supplementary Fig. 8). This result illustrates that EVCit linkers conjugated to a mAb are more sensitive to cathepsin B-mediated cleavage than VCit and SVCit linkers, which is consistent with the responsiveness of pyrene probes **1a**–**c** (Supplementary Fig. 2b). The ADCs were also tested for responsiveness to cathepsins L and S, which are also responsible for lysosomal cleavage of VCit linkers[34] (Supplementary Table 1). Interestingly, VCit ADC **3a** was slightly more sensitive to cathepsin L-mediated cleavage than EVCit **3c**. Cathepsin S cleaved both linker systems at similar rates. In addition, both sequences were cleaved at almost equivalent rates in a mixture of cathepsins B, L, and S. Thus, while highly responsive to cathepsin B-mediated cleavage, EVCit was not necessarily more sensitive than VCit to cleavage by other cathepsins.

Next, we assessed the ADCs for stability in human and mouse plasma. No significant degradation was observed in any of the ADCs after incubation in human plasma at 37 °C for 28 days (Fig. 3a). In contrast, although EVCit ADC **3c** showed almost no linker cleavage even after 14-day incubation in undiluted BALB/c mouse plasma, VCit and SVCit ADCs **3a**, **b** lost > 95% and ~70% of the conjugated MMAF after the same period of time (Fig. 3b and Supplementary Fig. 9). This tendency is consistent with what we observed in the small-molecule probes (Fig. 1c). The PEG spacer within the linker scaffold most likely facilitates the enzyme access to the linker–payload moiety. Considering this point, our results indicate that the acidic EVCit linker provides not only reactivity to cathepsin-mediated cleavage but also high stability in plasma despite a high degree of payload exposure.

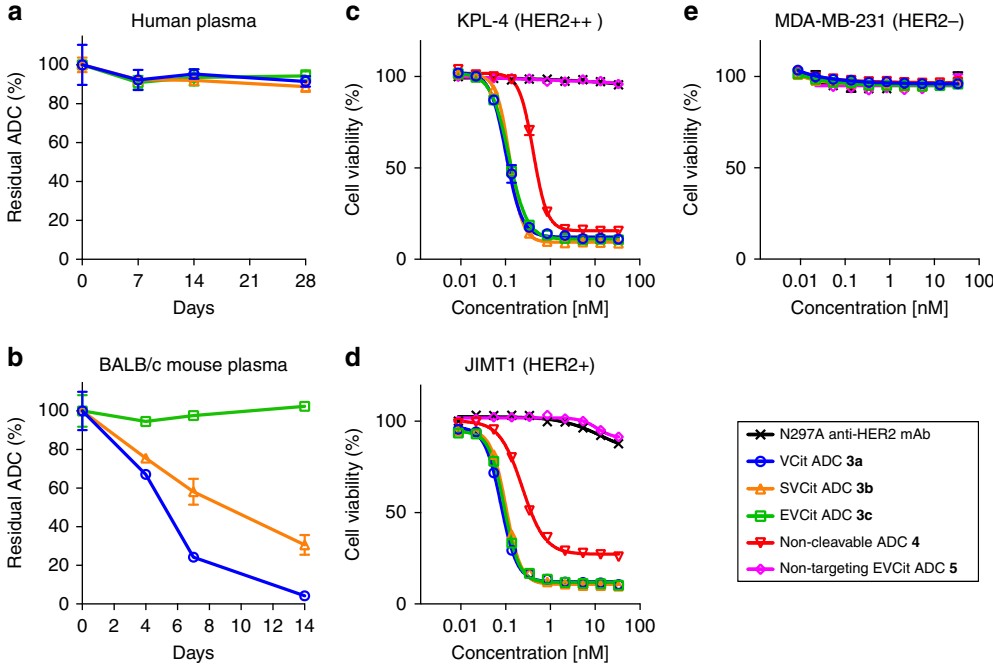

**Fig. 3** Plasma stability and in vitro cytotoxicity. **a** Stability in human plasma. **b** Stability in mouse plasma. Cell killing potency in the breast cancer cell lines KPL-4 (**c**), JIMT-1 (**d**), and MDA-MB-231 (**e**). We tested unconjugated N297A anti-HER2 mAb (black cross), VCit ADC (**3a**, blue circle), SVCit ADC (**3b**, orange triangle), EVCit ADC (**3c**, green square), non-cleavable ADC (**4**, red inversed triangle), and isotype control ADC containing EVCit (**5**, magenta diamond, non-targeting control). All assays were performed in quadruplicate. Error bars represent s.e.m

Subsequently, we tested the anti-HER2 ADCs for antigen binding affinity and specificity using the human breast cancer cell lines KPL-4 (HER2 positive) and MDA-MB-231 (HER2 negative) (Supplementary Fig. 10 and Supplementary Table 2). Like the parental anti-HER2 mAb, ADCs **3a–c** showed high binding affinity to KPL-4 ($K_D$: 0.12–0.16 nM) but not to MDA-MB-231. Non-targeting control **5** showed no binding to either cell line. These results indicate that installation of glutamic acid or serine at the $P_3$ position is unlikely to drastically influence the antigen recognition and specificity. We also evaluated these ADCs for in vitro cytotoxicity using HER2 positive- (KPL-4, JIMT-1, BT-474, and SKBR-3) and negative (MDA-MB-231) breast cancer cell lines (Fig. 3c–e, Supplementary Fig. 11, and Supplementary Tables 3, 4). Cathepsin-cleavable ADCs **3a–c** exhibited subnanomolar-level cell killing potency in the HER2-positive cell lines, but no cytotoxicity in HER2-negative MDA-MB-231 under our assay conditions. We did not observe significant difference in cell killing potency among the three cleavable ADCs (ranges of the $EC_{50}$ values in KPL-4: 0.10–0.12 nM; in JIMT-1: 0.078–0.10 nM; in BT-474: 0.058–0.063 nM; and in SKBR-3: 0.27–0.34 nM, Supplementary Table 3). Given that multiple proteases including cathepsins are most likely responsible for lysosomal processing of VCit-type linkers[34], there might be no distinct difference in the rate of lysosomal cleavage between the VCit and EVCit sequences. This speculation is also supported by our observation that both VCit and EVCit linkers were processed at almost equal rates in the presence of multiple cathepsins (Supplementary Table 1). However, non-cleavable ADC **4**, which lacks a cathepsin-cleavable sequence within the linker scaffold, showed 1.8–4.2-fold higher $EC_{50}$ values than those of cleavable ADC **3a** (Fig. 3c, d, Supplementary Fig. 11, and Supplementary Table 3). These results suggest that existence of a cleavage mechanism is a key to maximize cell killing potency of ADCs constructed using our branched linker platform.

**Validation of ADCs in vivo.** Finally, we evaluated the ADCs in vivo using mouse models. We first assessed PK profiles of VCit,

SVCit, and EVCit ADCs **3a–c** using BALB/c mice. Mice were treated with intravenous injection of each ADC or the parental N297A anti-HER2 mAb (3 mg kg$^{-1}$). Blood was collected periodically via the tail vein. Concentrations of total mAb (both conjugated and unconjugated) and intact ADC (conjugated only) in blood were determined by sandwich enzyme-linked immunosorbent assay (ELISA, Fig. 4a, b, Supplementary Fig. 12, and Supplementary Table 5). All ADCs showed similar clearance rates as that of the parental mAb ($t_{1/2\beta} = 14.9$ days), indicating that installing glutamic acid or serine at the $P_3$ position did not negatively impact the clearance profile (Fig. 4a). As expected, EVCit ADC **3c** showed no significant loss of payload caused by cleavage in circulation ($t_{1/2\beta} = 12.0$ days). In contrast, VCit and SVCit ADCs **3a, b** quickly lost MMAF ($t_{1/2\beta} = 2.0$ days and 2.4 days, respectively), demonstrating that the VCit and SVCit sequences installed on the mAb–branched linker system were unstable in circulation.

Encouraged by this finding, we tested VCit and EVCit ADCs **3a, c** for in vivo treatment efficacy in JIMT-1 and KPL-4 xenograft mouse models (Fig. 4c–f and Supplementary Fig. 13, 14). It has been reported that athymic nude mice quickly clear exogenously introduced IgGs[35]. Therefore, to prevent fast clearance of administered ADCs, tumor-bearing mice were preconditioned by intravenous administration of human IgGs (30 mg kg$^{-1}$)[36, 37]. A single dose of each ADC (1 or 3 mg kg$^{-1}$) or vehicle control was injected intravenously into tumor-bearing mice. Tumor volume and body weight were measured every 3 days. No significant toxicity caused by administration of either ADC was observed over the course of study (Supplementary Fig. 14). A single dose of EVCit-based ADC **3c** at 3 mg kg$^{-1}$ was curative and no tumor regrowth was visually observed in either model at the end of study (Fig. 4c–f). Furthermore, ADC **3c** was potent even at a lower dose (1 mg kg$^{-1}$) in the JIMT-1 model and all five mice that received this treatment survived over the course of study (Fig. 4c, e). In contrast, VCit ADC **3a** exhibited only partial inhibition of tumor growth despite the high in vitro cell

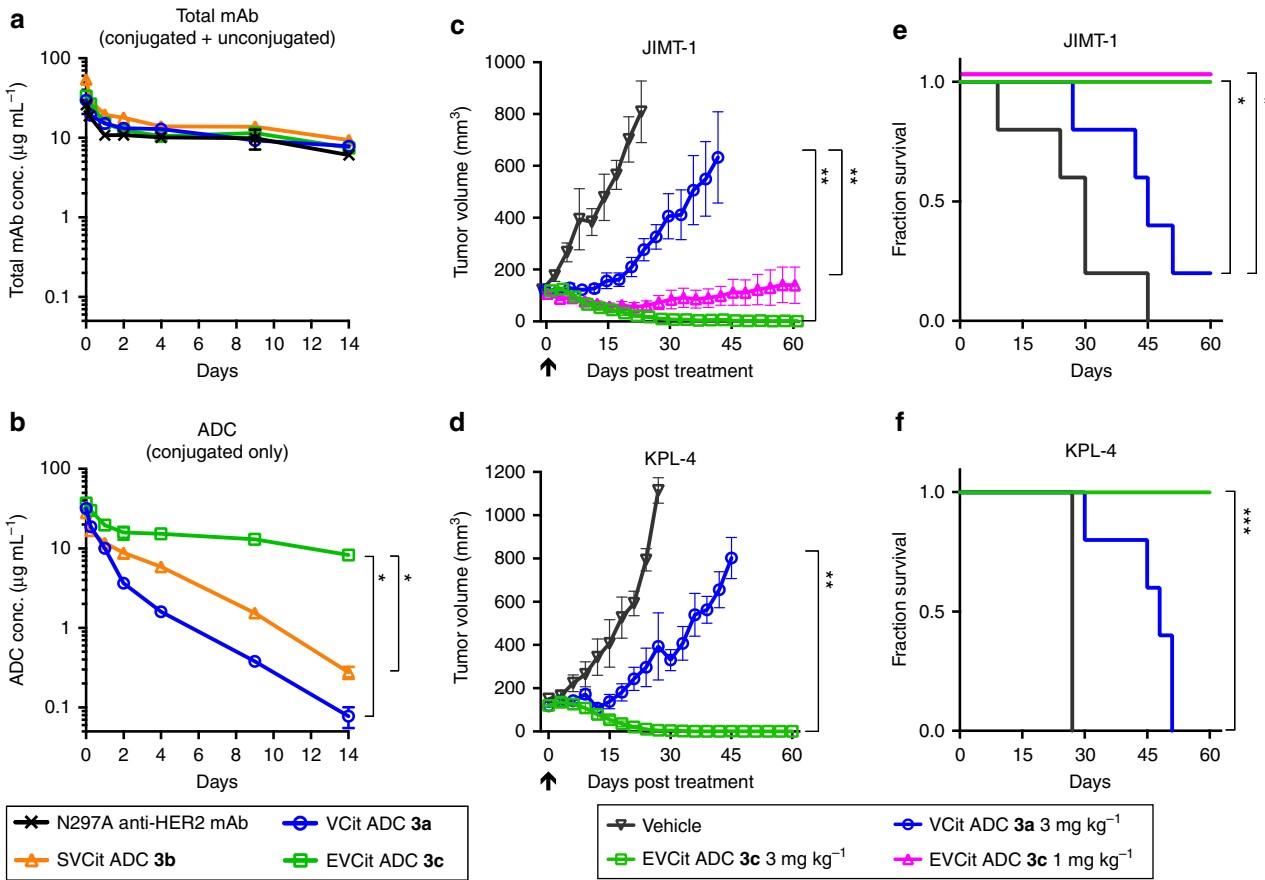

**Fig. 4** In vivo pharmacokinetics (PK) and antitumor activity. **a, b** PK of unconjugated N297A anti-HER2 mAb (black cross), VCit (blue circle), SVCit (orange triangle), and EVCit (green square) ADCs (**3a–c**) in female BALB/c mice ($n = 3$). At the indicated time points, blood was collected to quantify total antibody (conjugated and unconjugated, **a**) and ADC (conjugated only, **b**) by sandwich ELISA. **c, d** Antitumor activity of anti-HER2 ADCs (**3a**, **c**) in the JIMT-1 (**c**) and KPL-4 (**d**) xenograft tumor models (female NCr nude mice, $n = 3$ for vehicle in the KPL-4 model; $n = 5$ for vehicle in the JIMT-1 model and ADCs in both models). A single dose of VCit ADC (**3a**, 3 mg kg$^{-1}$, blue circle), EVCit ADC (**3c**, 3 mg kg$^{-1}$, green square; 1 mg kg$^{-1}$, magenta triangle), or vehicle control (gray inversed triangle) was administered to mice when a mean tumor volume reached ~100 mm$^3$ (indicated with a black arrow). Error bars represent s.e.m. **e, f** Changes in the percentages of surviving mice over time in the JIMT-1 (**e**) and KPL-4 (**f**) models. The curve of ADC (**3c**) at 1 mg kg$^{-1}$ (magenta, **e**) is slightly shifted upward for clarity. Mice were euthanized at the pre-defined endpoint (see the Method). * $P < 0.025$, ** $P < 0.01$, *** $P < 0.005$ (PK analysis: Welch's $t$ test; tumor volume on Day 27: Mann–Whitney $U$ test; survival curve: log rank test). The vehicle control groups were not used for statistical analysis

killing potency. Almost all mice that received this treatment died or reached a humane endpoint that required euthanasia before the end of study (four out of five mice dead in the JIMT-1 model; all five mice dead in the KPL-4 model) (Fig. 4c–f). Taking into account the molecular structure of ADC **3c**, these results demonstrate that the EVCit cleavable linker system can fully elicit the therapeutic potential of ADCs in mouse models even if it is spatially sequestered from the mAb through a long spacer.

## Discussion
We have shown that VCit-containing acidic tripeptides with high polarity, in particular an EVCit tripeptide sequence, have significantly enhanced stability in mouse and human plasma while remaining susceptible to intracellular cathepsin-mediated proteolytic cleavage. Notably, the small molecule-based stability assay clearly demonstrates that a carboxylic acid side chain at the P$_3$ position provides much greater stabilization effect than does a 2-hydroxyacetamide group, the modifier that reportedly conferred the VCit sequence with the highest stability in mouse plasma[23]. These features make the EVCit sequence ideal cleavable ADC linker design for increasing the hydrophilicity under physiological

conditions, maximizing the therapeutic potential, and minimizing the risk of systemic toxicity in mouse models caused by premature payload release. Indeed, a homogeneous anti-HER2 ADC constructed using an EVCit–PABC linker along with our branched linker technology[29] exhibited higher hydrophilicity and by far greater long-term in vivo stability than did ADCs equipped with a conventional VCit or SVCit, an analogue of the hydroxy-functionalized tripeptide ADC linker that reportedly exhibited increased stability in mouse plasma[23]. In addition, although treatment with a VCit-based anti-HER2 ADC showed poor therapeutic effect, treatment with the stable EVCit ADC led to complete remission in two xenograft mouse models of HER2-positive breast cancer. Both ADCs contain long PEG spacers within the linker scaffold fully exposing the cleavable peptide moieties. Thus, the EVCit linker system most likely provides great resistance to Ces1c-mediated degradation in mouse models even with a high degree of exposure. Although the EVCit sequence is promising in its present form, future structure–activity relationship studies on the interaction between this peptide sequence and the mouse Ces1c will provide in-depth insights into the observed stabilizing effect. Such understanding may enable us to design further improved ADC linkers.

The use of EVCit or similar peptide linkers (e.g., EVA, DVCit, DVA) could serve as a simple but powerful solution to salvage many types of ADCs previously abandoned due to linker instability in mouse models. The high polarity of the EVCit linker could also help mitigate the aggregation and fast clearance issues associated with hydrophobic high-DAR ADCs[38]. In addition, EVCit linkers may be preferentially chosen over non-cleavable linkers in the future design of various ADCs. Non-cleavable linkers are designed to withstand proteolytic degradation in circulation. They have been successfully used for constructing potent ADCs along with MMAF[39], monomethyl auristatin D (MMAD)[40], and emtansine (DM1)[7, 8]. However, the use of non-cleavable linkers reportedly attenuates or nullifies ADC potency of some payload molecules including doxorubicin[39], monomethyl auristatin E (MMAE)[34, 39], a hydrophilic auristatin derivative[41], and a pyrrolobenzodiazepine dimer (PBD)[34]. Attenuation of ADC potency arises because non-cleavable linkers lack a defined cleavage mechanism; after intracellular protein degradation, final active metabolites retain the linker component. It has been demonstrated that this drawback can be circumvented in some cases by fine-tuning the chemical structures of the linker and payload[42, 43]. However, the success of such efforts depends on the choice of the linker installation sites, conjugation modality, and payload. Indeed, as we have demonstrated in this and previous reports[29], our branched linker technology requires both adequate spacers and cleavable mechanisms within the linker scaffold for maximal ADC potency. These components are critical to alleviate the structural congestion and to ensure rapid payload release in an active form from each linker arm.

In summary, our findings support the conclusion that the EVCit linker technology is a significant contribution to efforts for developing next-generation ADCs and other drug conjugates. This technology will allow for flexible molecular design by minimizing challenges of linker instability and poor potency in preclinical studies. With further validation and optimization, this linker technology will benefit a diverse array of conjugation methods and linker systems developed to date, including conventional couplings at lysine or cysteine residues, site-specific conjugations at solvent accessible moieties (e.g., conjugation at the C-terminus of the antibody heavy chain)[44–46], and branched ADC linkers for heterologous payload loading[47, 48]. Furthermore, the long half-life of the acidic tripeptide linkers, as seen for EVCit and DVCit probes **1c, d**, will be useful for constructing small molecule-based drug conjugates for targeted therapy[49, 50].

## Methods

**Compounds**. Synthesis details and characterization data of all compounds in this study are described in the Supplementary Method section and Supplementary Fig. 15–38.

**Cathepsin B-mediated cleavage assay using pyrene probes**. Each test compound (10 mM in DMSO, 2 μL) was mixed with 97 μL of MES buffer (25 mM MES-Na, 1 mM DTT, pH 5.0) and 1 μL of 1-pyrenemethylamine (10 mM in DMSO, internal standard). The mixture was incubated at 37 °C for 10 min. Prewarmed human liver cathepsin B (20 ng μL$^{-1}$, EMD Millipore) in 100 μL MES buffer was added to the mixture, followed by incubation at 37 °C. Aliquots (10 μL) were collected at each time point (0, 0.5, 1, 3, 24, and 48 h). Cold acetonitrile containing 1% formic acid (40 μL) was added to precipitate proteins. Precipitated proteins were separated by centrifugation (15,000 × $g$, 4 °C, 30 min) and supernatant of each sample was analyzed for quantification by analytical HPLC (UV absorption at 342 nm). The amount of each probe was normalized to the peak area of the internal standard. All assays were performed at least three times in technical duplicate, and data shown are representative of the replicates.

**Plasma stability test using pyrene probes**. Each test compound (10 mM in DMSO, 2 μL) was mixed with 1 μL of 1-pyrenemethylamine (10 mM in DMSO, internal standard) and incubated at 37 °C for 10 min. Pre-warmed human plasma pooled from healthy donors or BALB/c mouse plasma (197 μL, Innovative Research) was added to the mixture, followed by incubation at 37 °C. Aliquots

(10 μL) were collected at each time point (0, 1, 6, 24, 48, and 96 h) and 40 μL of cold acetonitrile containing 1% formic acid was added to precipitate proteins. Supernatant of each sample was obtained and analyzed for quantification by analytical HPLC as described above. All assays were performed in triplicate.

**MTGase-mediated antibody–linker conjugation**. See the Supplementary Information for the preparation of human mAbs with a N297A mutation. Anti-HER2 IgG1 with a N297A mutation (291 μL in PBS, 11.79 mg mL$^{-1}$, 3.43 mg antibody) was incubated with branched linker **2** (18.3 μL of 100 mM stock in water, 80 equiv.) and Activa TI® (77 μL of 40% solution in PBS, Ajinomoto, purchased from Modernist Pantry) at room temperature for 16–20 h. The reaction was monitored using an Agilent G1946D LC/ESI-MS system equipped with a MabPac RP column (3 × 50 mm, 4 μm, Thermo Scientific). Elution conditions were as follows: mobile phase $A$ = water (0.1% formic acid); mobile phase $B$ = acetonitrile (0.1% formic acid); gradient over 6.8 min from $A:B$ = 75:25 to 1:99; flow rate = 0.4 mL min$^{-1}$. The conjugated antibody was purified by SEC (Superdex 200 increase 10/300 GL, GE Healthcare, solvent: PBS, flow rate = 0.6 mL min$^{-1}$) to afford an antibody-linker conjugate (3.15 mg, 92% yield determined by bicinchoninic acid assay).

**Strain-promoted azide–alkyne cycloaddition for payload installation**. DBCO–VCit–PABC–MMAF (12.6 μL of 4 mM stock solution in DMSO, 1.5 equivalent per azide group) was added to a solution of the mAb–linker conjugate in PBS (303.0 μL, 4.0 mg mL$^{-1}$), and the mixture was incubated at room temperature for 1 h. The reaction was monitored using an Agilent G1946D LC/ESI-MS system equipped with a MabPac RP column and the crude products were purified by SEC to afford ADC **3a** (>95% yield determined by bicinchoninic acid assay). Analysis and purification conditions were the same as described above (see the previous section). Average DAR values were determined based on UV peak areas. ADCs **3b, c** were prepared in the same manner. Purified ADCs were formulated in citrate buffer (20 mM sodium citrate and 1 mM citric acid, pH 6.6) containing 0.1% Tween 80 and trehalose (70 mg mL$^{-1}$)[32] and stored at 4 °C.

**HIC analysis**. Each ADC (1 mg mL$^{-1}$, 10 μL in PBS) was analyzed using an Agilent 1100 HPLC system equipped with a MAbPac HIC-Butyl column (4.6 × 100 mm, 5 μm, Thermo Scientific). Elution conditions were as follows: mobile phase $A$ = 50 mM sodium phosphate containing ammonium sulfate (1.5 M) and 5% isopropanol (pH 7.4); mobile phase $B$ = 50 mM sodium phosphate containing 20% isopropanol (pH 7.4); gradient over 30 min from $A:B$ = 99:1 to 1:99; flow rate = 0.5 mL min$^{-1}$.

**Long-term stability test**. Each ADC (1 mg mL$^{-1}$, 100 μL) in PBS was incubated at 37 °C. Aliquots (10 μL) were taken at each time point (7, 14, and 28 days) and immediately stored at −80 °C until use. Samples were analyzed using an Agilent 1100 HPLC system equipped with a MAbPac SEC-1 analytical column (4.0 × 300 mm, 5 μm, Thermo Scientific). Elution conditions were as follows: flow rate = 0.2 mL min$^{-1}$; solvent = PBS.

**Antibodies for ELISA**. All antibodies used in the ELISA assays in this study were purchased from commercial vendors as follows: Rabbit anti-MMAF antibody (LEV-PAF1) from Levena Biopharma, goat anti-human IgG Fab–HRP conjugate (109-035-097), goat anti-human IgG Fc antibody (109-005-098), and donkey anti-human IgG–HRP conjugate (709-035-149) from Jackson ImmunoResearch, and goat anti-rabbit IgG–HRP conjugate (32260) from Thermo Scientific.

**Plasma stability test using ADCs**. [1] Stability in mouse plasma. Each ADC (100 μg mL$^{-1}$, 1.2 μL in PBS) was added to undiluted BALB/c mouse plasma (118.8 μL) to a final concentration of 1 μg mL$^{-1}$. After incubation at 37 °C for varying time, aliquots (15 μL each) were taken and stored at −80 °C until use. Samples were analyzed by sandwich ELISA assay. A high-binding 96-well plate (Corning) was coated with rabbit anti-MMAF antibody (100 ng per well). After overnight coating at 4 °C, the plate was blocked with 100 μL of 2% BSA in PBS containing 0.05% Tween 20 (PBS-T) with agitation at room temperature for 1 h. Subsequently, the solution was removed and each ADC sample (100 μL in PBS-T containing 1% BSA) was added to each well, and the plate was incubated at room temperature for 2 h. After each well was washed three times with 100 μL of PBS-T, 100 μL of goat anti-human IgG Fab–HRP conjugate (1:10,000) was added. After being incubated at room temperature for 1 h, the plate was washed three times with 100 μL of PBS-T and 100 μL of 3,3′,5,5′-tetramethylbenzidine (TMB) substrate (0.1 mg mL$^{-1}$) in phosphate–citrate buffer/30% H$_2$O$_2$ (1:0.0003 volume to volume, pH 5) was added. After color was developed for 10−30 min, 25 μL of 3$N$-HCl was added to each well and then the absorbance at 450 nm was recorded using a plate reader (Biotek Cytation 5). Concentrations were calculated based on a standard curve. [2] Stability in human plasma. Assays were performed in the same manner using human HER2 (100 ng per well, ACROBiosystems) for plate coating, rabbit anti-MMAF antibody (1:5,000) and goat anti-rabbit IgG–HRP conjugate (1:10,000) as secondary and tertiary detection antibodies, respectively. All assays were performed in triplicate.

**Human cathepsin cleavage assay for ADCs**. Each ADC (1 mg mL$^{-1}$) in 30 μL of MES buffer (10 mM MES-Na, 40 μM DTT, pH 5.0) was incubated at 37 °C for

10 min. To the solution was added pre-warmed human cathepsin B (20 ng μL$^{-1}$), cathepsin L (20 ng μL$^{-1}$), cathepsin S (2 ng μL$^{-1}$, all cathepsins from EMD Millipore), or a mixture of the three (B: 6.67 ng μL$^{-1}$; L: 6.67 ng μL$^{-1}$; S: 0.67 ng μL$^{-1}$) in 30 μL MES buffer, followed by incubation at 37 °C. Aliquots (20 μL) were collected at each time point (4, 10.5, and 24 h in Supplementary Fig. 8; 5 and 24 h in Supplementary Table 1) and treated with EDTA-free protease inhibitor cocktails (0.5 μL of 100X solution, Thermo Scientific). All samples were analyzed using an Agilent 1100 HPLC system equipped with a MabPac RP column (3 × 50 mm, 4 μm, Thermo Scientific). Elution conditions were as follows: Mobile phase A = water (0.1% formic acid; mobile phase B = acetonitrile (0.1% formic acid); gradient over 6.8 min from A:B = 75:25 to 1:99; flow rate = 0.4 mL min$^{-1}$. Average DAR values were determined based on UV peak areas.

**Cell culture.** JIMT-1 (AddexBio), BT-474 (ATCC), and SKBR-3 (ATCC) were cultured in RPMI1640 (Corning) supplemented with 10% EquaFETAL® (Atlas Biologicals), GlutaMAX® (2 mM, Gibco), sodium pyruvate (1 mM, Corning), and penicillin-streptomycin (penicillin: 100 units mL$^{-1}$; streptomycin: 100 μg mL$^{-1}$, Gibco). KPL-4 (provided by Dr. Junichi Kurebayashi at Kawasaki Medical School)[51] and MDA-MB-231 (ATCC) were cultured in DMEM (Corning) supplemented with 10% EquaFETAL®, GlutaMAX® (2 mM), and penicillin-streptomycin (penicillin: 100 units mL$^{-1}$; streptomycin: 100 μg mL$^{-1}$). All cells were cultured at 37 °C under 5% CO$_2$ and passaged before becoming fully confluent up to 10 passages. All cell lines were periodically tested for mycoplasma contamination. Cells were validated for the HER2 expression level in cell-based ELISA prior to use (see the following Cell-based ELISA section).

**Cell-based ELISA.** Cells (KPL-4 or MDA-MB-231) were seeded in a culture-treated 96-well clear plate (10,000 cells per well in 100 μL culture medium) and incubated at 37 °C under 5% CO$_2$ for 24 h. Paraformaldehyde (8%, 100 μL) was added to each well and incubated for 15 min at room temperature. The medium was aspirated and the cells were washed three times with 100 μL of PBS. Cells were treated with 100 μL of blocking buffer (0.2% BSA in PBS) with agitation at room temperature for 2 h. After the blocking buffer was discarded, serially diluted samples (in 100 μL PBS containing 0.1% BSA) were added and the plate was incubated overnight at 4 °C with agitation. The buffer was discarded and the cells were washed three times with 100 μL of PBS containing 0.25% Tween 20. Cells were then incubated with 100 μL of donkey anti-human IgG–HRP conjugate (diluted 1:10,000 in PBS containing 0.1% BSA) at room temperature for 1 h. The plate was washed three times with PBS containing 0.25% Tween 20, and 100 μL of TMB substrate (0.1 mg mL$^{-1}$) in phosphate–citrate buffer/ 30% H$_2$O$_2$ (1:0.0003 volume to volume, pH 5) was added. After color was developed for 10–30 min, 25 μL of 3$N$-HCl was added to each well and then the absorbance at 450 nm was recorded using a plate reader (Biotek Cytation 5). Concentrations were calculated based on a standard curve. $K_D$ values were then calculated using Graph Pad Prism 7 software. All assays were performed in triplicate.

**Cell viability assay.** Cells were seeded in a culture-treated 96-well clear plate (5,000 cells per well in 50 μL culture medium) and incubated at 37 °C under 5% CO$_2$ for 24 h. Serially diluted samples (50 μL) were added to each well and the plate was incubated at 37 °C for 72 h (KPL-4 and SKBR-3) or 96 h (JIMT-1, MDA-MB-231, and BT-474). After the old medium was replaced with 100 μL fresh medium, 20 μL of a mixture of WST-8 (1.5 mg mL$^{-1}$, Cayman chemical) and 1-methoxy-5-methylphenazinium methylsulfate (1-methoxy PMS, 100 μM, Cayman Chemical) was added to each well, and the plate was incubated at 37 °C for 2 h. After gently agitating the plate, the absorbance at 460 nm was recorded using a plate reader. EC$_{50}$ values were calculated using Graph Pad Prism 7 software. All assays were performed in quadruplicate.

**Animal experiments.** All procedures were approved by the Animal Welfare Committee of the University of Texas Health Science Center at Houston and performed in accordance with the institutional guidelines for animal care and use.

**In vivo pharmacokinetics study.** Female BALB/c mice (6−8 weeks old, $n$ = 3 per group, The Jackson Laboratory) were injected intravenously with each mAb or ADC sample (100 μL) at a dose of 3 mg kg$^{-1}$. After injection, blood (5−10 μL) was drawn through the tail vein at each time point (15 min, 6 h, 1 day, 2 days, 4 days, 9 days, and 14 days) and processed with 5 mM EDTA in PBS. Plasma samples were stored at −80 °C until use. All mice were humanely euthanized after last blood collection. Plasma samples were analyzed by sandwich ELISA. For determination of the total antibody concentration (both conjugated and unconjugated), goat anti-human IgG Fc antibody (500 ng per well) and goat anti-human IgG Fab–HRP conjugate (1:5,000) were used for plate coating and detection, respectively. For determination of ADC concentration (conjugated only), rabbit anti-MMAF antibody (100 ng per well) and goat anti-human IgG Fab–HRP conjugate (1:10,000) were used in the same manner. Assays were performed in the same manner as described above (see the section of the plasma stability test for ADCs). Concentrations were calculated based on a standard curve. Half-life at the elimination phase ($t_{1/2β}$) was estimated using methods for non-compartmental analysis[52] and area under the curve (AUC$_{0-14 \ days}$, h × μg mL$^{-1}$) of each sample was calculated using GraphPad Prism 7 software (Supplementary Table 5).

**In vivo xenograft mouse models of human breast cancer.** To produce KPL-4 or JIMT-1 tumors, female NCr nude mice (6−8 weeks old, Taconic Biosciences) were orthotopically injected into the mammary fat pad with 5−7 × 10$^6$ cells suspended in 50 μL of 1:1 PBS/Cultrex® BME Type 3 (Trevigen). When the tumor volume reached ~100 mm$^3$, mice were randomly assigned to three or four groups ($n$ = 3 for vehicle in both models; $n$ = 5 for vehicle in the JIMT-1 model and ADCs in both models) and preconditioned with sterile-filtered human IgG (30 mg kg$^{-1}$, Innovative Research, catalog number: IRHUGGFLY1G) injected via the tail vein[36, 37]. After 24 h, a single dose of 100 μL VCit ADC **3a** (3 mg kg$^{-1}$), EVCit ADC **3c** (1 or 3 mg kg$^{-1}$), or vehicle was administered to mice intravenously. Tumor volume and body weight were monitored every 3 days using a digital caliper. Mice were euthanized when the tumor volume exceeded 1000 mm$^3$, the tumor size exceeded 2 cm in diameter, greater than 15% weight loss was observed, or mice showed signs of distress.

**Data reporting.** No statistical analysis was performed to determine sample size prior to experiments, but the sample size was determined according to similar experiments in the field reported previously. We did not intend to use the vehicle control groups in the xenograft studies for statistical analysis. The investigators were not blinded to allocation during experiments. No samples or animals were excluded from the studies. For the in vivo PK analysis, a Welch's $t$-test (two-tailed, unpaired, uneven variance) was used to determine statistical significance of the observed differences. For the xenograft model studies, a Mann–Whitney $U$ test (two-tailed, unpaired, non-parametric) was used. Kaplan-Meier survival curve statistics were analyzed with a log-rank (Mantel–Cox) test. To adjust the family wise error rate with a Bonferroni correction, $P$ values less than [0.05 divided by the number of comparisons] were considered statistically significant. See Supplementary Table 6 for all $P$ values.

**Data availability.** The authors declare that all the data supporting the findings of this study are available within the paper, its supplementary information file, or from the corresponding author upon reasonable request.

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

## Acknowledgements

We gratefully acknowledge Prof. Junichi Kurebayashi (Kawasaki Medical School) for providing us with the cell line KPL-4. We thank Dr. Mohmmad Y. Wani for technical support and insightful opinions, and Dr. Georgina T. Salazar for editing the manuscript. This work was supported by the Department of Defense (the Breast Cancer Research Program, W81XWH-18-1-0004 to K.T.), The University of Texas System (Regents Health Research Scholars Award to K.T.), the Cancer Prevention and Research Institute of Texas (RP150551 to Z.A.), and the Welch Foundation (AU-0042-20030616 to Z.A.).

## Author contributions

Y.A. and C.M.Y. contributed equally to the work. K.T. conceived the project rationale and supervised all experiments. Y.A., C.M.Y., and K.T. designed experiments and wrote the manuscript. W.X., X.G., N.Z., and Z.A. produced mutated monoclonal antibodies. Y.A. and C.M.Y. prepared the probes, linkers, and payload components and performed the in vitro assays using the probes. Y.A. constructed and characterized all ADCs. Y.A. and C.M.Y. performed the cell killing assays. Y.A. and K.T. performed the pharmacokinetic studies. C.M.Y. performed the xenograft studies.

## Additional information

**Competing interests:** Y.A., C.M.Y., N.Z., Z.A., and K.T. are named inventors on a pending patent application relating to the work filed by The University of Texas Health Science Center at Houston. The remaining authors declare no competing interests.

