## [Peer Review File · Nature Communications]

Reviewers' comments:

Reviewer #1 (Remarks to the Author):

Reviewers comments

The manuscript by Anami et al. describes the development of modified linkers that help improve stability in mouse serum which, unlike humans and most other species, contains carboxylesterase 1c. By adding a glutamic acid residue next to valine-citrulline (P3 position), the authors nicely demonstrate the improved stability of the linker. As a bonus, the modified ADC is also slightly less hydrophobic which should also help with PK in vivo. The experiments are convincing, and the manuscript is very well written and advances the ADC field. My main concern is whether the advance is significant enough to warrant publication in Nature Communications. Some detailed concerns are as follows:

The previous manuscript by Dorywalska et al, (Mol Can Ther, 2016, PMID: 26944918) already describes the development of a modified Val-Cit linker that is much more stable in mouse serum. Unfortunately, the authors do not compare their new linkers with the previous one, so it is unclear if the new format is superior to the previous one. The authors use serine in the P3 position for comparison to the Dorywalska paper. However, serine lacks the key carbonyl group (carbon atom double-bonded to oxygen) that is present in the most effective linkers from the Dorywalska paper. The carbonyl group is also present in the authors most effective linkers: EVCit and DVCit. Therefore, the question remains, are the new linkers any more effective than the ones published in 2016?

The authors approach provides a new tool that can be used to help prevent premature cleavage of ADCs in mouse serum. However, alternative ways to mitigate the problems should be highlighted. For example, one could use carboxylesterase 1c knockout mice available from The Jackson Laboratory, or cross the Ces1c null alleles onto a nude mouse model. Susceptibility to carboxylesterase also depends on the site of drug attachment, and several conjugation sites are known to mitigate this issue.

Why are the EVCit 1c and DVCit 1d containing the surrogate payload (1-pyrenemethylamine) still relatively unstable in mouse plasma (~30% intact at 50h, figure 1C) compared to human plasma (~100% at 50h, supplemental figure 1a). Is this due to carboxylesterase 1c? With the MMAF-containing EVCit ADC 3c the stability in mouse serum appears to be much higher than it was with the surrogate (Fig 3b). Can the authors explain why the stability appears to be much better for the ADC?

The data from figure 4 support the authors conclusion that the modified EVCit ADC is much more potent than the traditional VCit ADC. However, the key with any new ADC modification is whether the therapeutic window has been improved. In other words, is the modified ADC also more toxic, or has efficacy been improved without increasing toxicity? Were body weights or other toxicity parameters measured in the tumor studies?

In figure 1 c, two linker structures show similar superior activity in mouse plasma – EVCit 1c and DVCit 1d. Are the small differences in half-life between these two statistically different? What was the reason for advancing EVCit and not DVCit?

The data in Figure 3a & b and supplemental figure 6a and 6b appear to be duplicated.

The legend in supplementary figure 6b is missing.

Reviewer #2 (Remarks to the Author):

"Glutamic acid-valine-citrulline cleavable linkers ensure high stability and efficacy of antibody-drug conjugates in mouse models" by Anami et al.

This paper addresses a very common problem in the ADC field. The valine-citrulline (Val-Cit) dipeptide, which is commonly used for cleavage of cytotoxic payloads from antibodies, is susceptible to premature digestion by an extracellular enzyme not found in cynomolgus monkey and human plasma. This instability in mouse plasma makes it challenging to evaluate Val-Cit based ADCs in preclinical mouse models. Anami et al. demonstrates greatly improved mouse plasma stability by placing an acidic residue at the N-terminus of the Val-Cit dipeptide. This work will enable preclinical analysis of a more diverse set of ADCs and is an important contribution to the ADC field.

Minor points

1. Will the authors please speculate as to why the serine modification did not show a great improvement in stability as they were expecting based on data from reference #23?

2. On pg 8, line 130, the authors state that the number of PEG units was adjusted so that all payload modules had similar lengths. It is unclear what this is referring to as it seems that all linkers use a PEG3 spacer. Could the authors please clarify? If different PEG spacer lengths are used, that could change interpretation of some of the data.

3. Due to the nature of the branched linker technology used and the somewhat unique requirements for a cleavable linker system, do the authors believe that their findings will necessarily be applicable to other linker systems where accessibility to the linker-payload is different?

4. There are minor typos on pg 3, line 25, pg 4, line 48 and pg 8, line 137.

Reviewer #3 (Remarks to the Author):

Tsuchikama and coworkers have written a very compelling manuscript describing a new variation on a classic ADC protease-cleavable linker that is stable to carboxylesterases. This improved stability results in improved pharmacokinetics as well as increased therapeutic efficacy. While this work was inspired by a publication from Dorywalska and coworkers at Pfizer to improve peptide linker stability, that work coupled improvements in peptide stability to site of conjugation and was limited in scope and applicability. Conversely, the impact of the current manuscript is high as it has broad utility. The EVCit linker is stable even in linker types that are likely highly solvent exposed, making it generally useful for a variety of conjugation, spacer, and payload types. This makes the work of high interest to the ADC field and also of interest to an even broader field of targeted delivery (both therapeutics and imaging agents). I think it is virtually guaranteed that this paper will influence the field and that EVCit will be the linker selected for many future test ADCs. An additional advantage of this linker is the increased hydrophilicity accompanying the incorporation of glutamic acid. This is supported by HIC data and hopefully in future work by examples of rescuing linker-drugs that fail due to aggregation as they have done here rescuing ADCs that fail due to poor PK caused by instability.

The paper was extremely well written and referenced. Statistical analysis was provided for all appropriate data including half-lives, cytotoxicity, stability (in vitro and in vivo), and efficacy.

The one suggestion I have is regarding the authors focus on cathepsin B as the protease primarily

responsible for cleavage of the peptide linker. Recent work (Ref. 40) indicates that other cysteine cathepsins are also responsible for this proteolytic activity. The authors indicate that cleavage by CatB of the different linkers is equivalent (Supp. Fig. 5). While this claim could be solidified by testing cleavage with more than one (or a mixture of cathepsins) or even a lysosomal fraction (commercial preps are available), their in vitro cleavage result is appropriately supported by ADC activity data. The authors should at least address the complexity of the issue of proteolytic cleavage in the text in that multiple proteases might ultimately be responsible.

Congratulations on a well-written, novel manuscript that in my opinion is of broad interest.

Thomas Pillow
Genentech

RESPONSE TO THE REVIEWERS

<Reviewer 1>

Overall Comment: The manuscript by Anami et al. describes the development of modified linkers that help improve stability in mouse serum which, unlike humans and most other species, contains carboxylesterase 1c. By adding a glutamic acid residue next to valine-citrulline (P₃ position), the authors nicely demonstrate the improved stability of the linker. As a bonus, the modified ADC is also slightly less hydrophobic which should also help with PK in vivo. The experiments are convincing, and the manuscript is very well written and advances the ADC field. My main concern is whether the advance is significant enough to warrant publication in Nature Communications. Some detailed concerns are as follows:

Reply to Overall Comment: We thank the reviewer for careful and rigorous review of this manuscript and for the insightful comments and constructive suggestions, which greatly helped further strengthen the impact of this manuscript. Our detailed response to the comments and concerns raised by the Reviewer follows hereunder. *Major changes/additions to the manuscript and Supplementary Information are highlighted in blue.*

Comment 1: The previous manuscript by Dorywalska et al, (Mol Can Ther, 2016, PMID: 26944918) already describes the development of a modified Val-Cit linker that is much more stable in mouse serum. Unfortunately, the authors do not compare their new linkers with the previous one, so it is unclear if the new format is superior to the previous one. The authors use serine in the P₃ position for comparison to the Dorywalska paper. However, serine lacks the key carbonyl group (carbon atom double-bonded to oxygen) that is present in the most effective linkers from the Dorywalska paper. The carbonyl group is also present in the authors most effective linkers: EVCit and DVCit. Therefore, the question remains, are the new linkers any more effective than the ones published in 2016?

Reply 1-1: We appreciate the reviewer's astute comment on the significance of our work in the field. We agree that we can undoubtedly demonstrate the degree of significance and novelty by directly comparing the EVCit linker with a linker containing almost the same chemical structure as reported in the paper by Dorywalska et al. To this end, we prepared a tripeptide probe containing a 2-hydroxyacetamide moiety at the P₃ position (probe **1f**, see updated Fig. 1b and Supplementary Scheme S1). The previous paper reported that this structure provided ADC linkers with the highest stability in mouse plasma. Small molecule-based probes are generally much more sensitive to enzymatic cleavage than linkers conjugated to an antibody. So, we expected that testing at the small-molecule level would provide stability data under harsh assay conditions. Please note that we performed a new set of stability test for all probes **1a-f** in triplicate to ensure the fidelity and to exclude any potential biases to the results based on differences in lots of assay medium, reagents, and mouse plasma. We have updated Fig. 1c and related descriptions in the manuscript based on this new result.

2-Hydroxyacetamide probe **1f** was slightly more stable in undiluted BALB/c mouse plasma than VCit and SVCit probes **1a,b** ($t_{1/2}$ = 5.0, 2.3, and 3.1 h, respectively) but much less stable than acidic EVCit and DVCit probes **1c,d** ($t_{1/2}$ = 19.1 and 14.0 h, respectively). The stability trend was consistent between this and previous experiments. Thus, this result strongly supports the following conclusion: 1) while somewhat effective, a neutral carbonyl group at the P₃ position of a modified Val-Cit linker is unlikely to significantly enhance linker stability in mouse plasma even in combination with a 2-hydroxy group; 2) the remarkable improvement of plasma stability observed in the EVCit and DVCit sequences

is derived from the negatively charged carboxylic acid side chain at the P₃ position rather than simply from the carbonyl moiety; 3) EVCit and DVCit are most likely more stable than any of the tripeptide sequences reported by Dorywalska et al. To describe this new finding, we have added the following sentences to the manuscript:

In the Results Section

“In particular, the 2-hydroxyacetamide group within probe **1f** is the modifier that provided the greatest stability in their report. Thus, we expected that comparing newly developed peptide sequences with probe **1f** would clearly demonstrate the degree of improvement over the previous linker design.”

“2-Hydroxyacetamide probe **1f** was slightly more stable ($t_{1/2} = 5.0$ h) than SVCit probe **1b** but much less stable than EVCit and DVCit probes **1c,d**. Thus, the stabilizing effect of a neutral carbonyl group at the P₃ position was not as significant as that of a negatively charged carboxylic acid side chain.”

In the Discussion Section

“Notably, the small molecule-based stability assay clearly demonstrates that a carboxylic acid side chain at the P₃ position provides much greater stabilization effect than does a 2-hydroxyacetamide group, the modifier that reportedly conferred the VCit sequence with the highest stability in mouse plasma.²³”

Note: The half-lives of probes **1a-e** determined in this experiment were shorter than those reported in the original manuscript. This may be because of the use of reagents and mouse plasma from different batches. The relatively large reduction of half-lives of EVCit and DVCit probes **1c,d** observed this time may also indicate that there were technical errors in the previous experiment (e.g., substrate or plasma sample prepared at incorrect concentrations). We repeated a new set of assays for all probes under the same conditions and confirmed reproducibility. Thus, we decided to update all stability data in Fig. 1c based on this new result. This update will not change our conclusion on the stabilization effect by modification at the P₃ position.

Reply 1-2: Testing a 2-hydroxyacetamide-functionalized ADC for plasma stability and pharmacokinetics would further highlight how advantageous the EVCit linker is in ADC construction. While interesting, we think such experiments are dispensable in supporting our conclusion of this manuscript. The above-mentioned experiment showed that the difference in half-life between SVCit probe **1b** and 2-hydroxyacetamide probe **1f** was only less than 2-fold. In addition, as described in Supplementary Fig. 6b, an ADC containing a 2-hydroxyacetamide-functionalized linker (the most stable ADC in the paper by Dorywalska et al) retained about 84% payload conjugated at a highly solvent-accessible site after 4.5-day incubation in mouse plasma. Given that our SVCit-ADC **3b** showed slightly lower but similar stability (75.6% payload retention after 4-day incubation), the SVCit ADC served in this study as a surrogate of the previously reported ADC with a similar stability profile. Taking into account these points, we speculate that 2-hydroxyacetamide-functionalized ADCs are highly unlikely to reach the same level of stability in mouse plasma as by EVCit ADC **3c**.

Comment 2-1: The authors approach provides a new tool that can be used to help prevent premature cleavage of ADCs in mouse serum. However, alternative ways to mitigate the problems should be highlighted. For example, one could use carboxylesterase 1c knockout mice available from The Jackson Laboratory, or cross the Ces1c null alleles onto a nude mouse model.

Reply 2-1: Thank you for suggesting alternative approaches to circumventing the premature cleavage of VCit-type linkers. We agree such approaches help circumvent the instability issue. However, we humbly

suggest that the use of such genetically engineered mice may hamper smooth implementation of in vivo studies because of long lead time, limited choice of parent genetic background, and relatively high sales price. Thus, we have added to the Introduction Section the description below:

“One approach to circumvent this problem is to use Ces1c-knockout mice (available from The Jackson Laboratory) or cross the Ces1c null alleles onto an immunocompromised mouse model. However, the use of such genetically engineered mice may hamper smooth implementation of in vivo studies because of long lead time and limited choice of parent genetic background.”

Comment 2-2: Susceptibility to carboxylesterase also depends on the site of drug attachment, and several conjugation sites are known to mitigate this issue.

Reply 2-2: We agree with the reviewer’s comment regarding the effect of the linker conjugation site on stability in plasma. We stated this point in the Introduction Section in the original manuscript as below:

“The linker instability in mouse plasma can also be ameliorated by carefully selecting the linker attachment sites within an antibody and limiting the length of the VCit linker to minimize the exposure of the vulnerable moiety to extracellular enzymes, as demonstrated with several VCit-based ADCs²³⁻²⁵.”

Comment 3-1: Why are the EVCit 1c and DVCit 1d containing the surrogate payload (1-pyrenemethylamine) still relatively unstable in mouse plasma (~30% intact at 50h, figure 1C) compared to human plasma (~100% at 50h, supplemental figure 1a). Is this due to carboxylesterase 1c?

Reply 3-1: Yes, human plasma lacks Ces1c that cleaves a VCit sequence. That is why all probes showed high stability in human plasma whereas varying degradation rates were observed in mouse plasma depending on the modification at the P₃ position.

Comment 3-2: With the MMAF-containing EVCit ADC 3c the stability in mouse serum appears to be much higher than it was with the surrogate (Fig 3b). Can the authors explain why the stability appears to be much better for the ADC?

Reply 3-2: This is a good question. In general, linkers conjugated to a protein such as a mAb are protected by a shield effect preventing the access of proteases such as Ces1c. As you pointed out above, this effect depends on the degree of linker exposure that is determined by the conjugation site and length of linker (the more exposed, the less protected). Even though EVCit ADC **3c** has relatively long spacers, we believe that the EVCit moiety within its scaffold still receives a certain level of protection. In contrast, there is no such shield mechanism with the small molecule-based probes containing 1-pyrenemethylamine. We believe this is why “naked” EVCit and DVCit probes **1c,d** were still responsive to Ces1c degradation while EVCit ADC **3c** showed almost complete resistance against degradation.

Comment 4: The data from figure 4 support the authors conclusion that the modified EVCit ADC is much more potent than the traditional VCit ADC. However, the key with any new ADC modification is whether the therapeutic window has been improved. In other words, is the modified ADC also more toxic, or has efficacy been improved without increasing toxicity? Were body weights or other toxicity parameters measured in the tumor studies?

Reply 4: We appreciate the reviewer's comment on the therapeutic window, a very important parameter in ADC research. Our EVCit ADC showed improved treatment efficacy without any toxicity in tumor mouse models. We have added to SI a figure depicting change of body weight during the treatment (Supplementary Fig. 11). No toxicity associated to the treatment (acute loss of body weight) was observed in either VCit- or EVCit ADC group. This absence of toxicity is probably due to the relatively high maximum tolerated dose (MTD) of MMAF (>16 mg/kg). We have also added to the Results section in the manuscript the sentence below:

“No significant toxicity caused by administration of either ADC was observed over the course of study (Supplementary Fig. 11).”

Although we feel it is beyond the scope of this manuscript, testing VCit and EVCit ADCs containing with more toxic payload than MMAF, such as MMAE (MTD: 1 mg/kg, see Page 42 of the attached patent application US8512707B2), will further validate the toxicity profile of EVCit ADCs. These experiments are currently underway in our laboratory.

Comment 5: In figure 1 c, two linker structures show similar superior activity in mouse plasma – EVCit 1c and DVCit 1d. Are the small differences in half-life between these two statistically different? What was the reason for advancing EVCit and not DVCit?

Reply 5: As mentioned in Reply 1-1, we have updated Fig 1c and the half-lives of EVCit and DVCit probes **1c,d** in mouse plasma were 19.1 h (95% CI: 16.3–22.6 h) and 14.0 h (95% CI: 13.3–14.7 h), respectively. The difference between these values is statistically significant (analyzed using GraphPad Prism 7.0), indicating that EVCit is superior to DVCit in terms of stability in mouse plasma. To reflect this new finding, we have added the following sentence to the Results section:

“The difference between these values is statistically significant as analyzed using GraphPad Prism 7.0. This result indicates that EVCit is superior to DVCit in terms of stability in mouse plasma.”

Another factor we took into account is aspartimide formation, which occasionally occurs as a side reaction during the synthesis of aspartic acid-containing peptides. Glutamic acid-containing peptides can also undergo such a side reaction but to a much lesser extent. Thus, we expected that the use of EVCit rather than DVCit would reduce potential risks in future production.

Comment 6: The data in Figure 3a & b and supplemental figure 6a and 6b appear to be duplicated.

Reply 6: Thank you for pointing out the duplicated figures. We intentionally did so in Supplementary Fig. 6 and 8 so that readers could easily figure out the data in both graph and table formats in the same page. For clarity, we have added the explanation “for ease of comparison, graph(s) shown in the manuscript are duplicated here” to the figure legends for Supplementary Fig. 6 and 8. If showing identical data both in the manuscript and SI is against Nature Communications' editorial policy, we will not hesitate to remove the duplicated graphs from the SI.

Comment 7: The legend in supplementary figure 6b is missing.

Reply 7: The figure legend starts with “Stability of ADCs **3a-c** in (a) human plasma and (b) undiluted BALB/c mouse plasma at 37 °C...”. We feel this is a sufficient description for both Supplementary Fig. 6a and 6b.

[End of response to Reviewer 1]

<Reviewer 2>

Overall Comment: “Glutamic acid-valine-citrulline cleavable linkers ensure high stability and efficacy of antibody-drug conjugates in mouse models” by Anami et al.

This paper addresses a very common problem in the ADC field. The valine-citrulline (Val-Cit) dipeptide, which is commonly used for cleavage of cytotoxic payloads from antibodies, is susceptible to premature digestion by an extracellular enzyme not found in cynomolgus monkey and human plasma. This instability in mouse plasma makes it challenging to evaluate Val-Cit based ADCs in preclinical mouse models. Anami et al. demonstrates greatly improved mouse plasma stability by placing an acidic residue at the N-terminus of the Val-Cit dipeptide. This work will enable preclinical analysis of a more diverse set of ADCs and is an important contribution to the ADC field.

Reply to Overall Comment: We thank the reviewer for taking time to review this manuscript. We appreciate the reviewer’s accurate summary and positive impression of our work. We believe this reviewer will more clearly recognize the significance of our work based on our response to this and the other reviewers and support acceptance of this revised manuscript for publication in *Nature Communications*.

Comment 1: Will the authors please speculate as to why the serine modification did not show a great improvement in stability as they were expecting based on data from reference #23?

Reply 1: This is a good question. As stated in Reply 3-2 to Reviewer 1, the small molecule-based probes are not conjugated to a protein such as a mAb and do not receive any shield effect. As a result, they have high sensitivity to many, if not all, proteases including carboxyesterase 1c. In other words, testing in the small molecule format allows for assessment of plasma stability purely on a peptide sequence of interest rather than as a part of a protein conjugate. We speculate this is a reason for the marginal improvement in stability as seen for serine-modified probe **1b**. In contrast, the ADC format provides the linker moiety with a shield effect, which we believe enabled SVCit ADC **3b** to exhibit moderately improved stability in mouse plasma (Fig. 3b and Supplementary Fig. 6b). Indeed, the most stable tripeptide linker in reference #23 was tested not in the small-molecule format but in the ADC format and showed stability comparable to what we observed in SVCit ADC **3b**.

Comment 2: On pg 8, line 130, the authors state that the number of PEG units was adjusted so that all payload modules had similar lengths. It is unclear what this is referring to as it seems that all linkers use a PEG3 spacer. Could the authors please clarify? If different PEG spacer lengths are used, that could change interpretation of some of the data.

Reply 2: We thank the reviewer for the helpful suggestion to clarify the description of the adjustment of linker length. We used a commercially available DBCO-**PEG4**-VCit-PABC-MMAF to construct VCit ADC **3a**. On the other hand, we used a **PEG3** unit instead of PEG4 to construct SVCit- and EVCit-payload modules for SVCit and EVCit ADC **3b,c** because they have one more α -amino acid residue within the linker sequence (serine and glutamic acid). To clarify this point, we have added the description below to the Results section:

“(PEG4 for the dipeptide VCit and PEG3 for the tripeptides SVCit and EVCit)”

Comment 3: Due to the nature of the branched linker technology used and the somewhat unique requirements for a cleavable linker system, do the authors believe that their findings will necessarily be applicable to other linker systems where accessibility to the linker-payload is different?

Reply 3: Yes, we think our findings will be broadly applicable to other linker systems and the use of EVCit linkers will provide multiple benefits. Reviewer 3 also acknowledges this point (see Overall Comment from Reviewer 3). We used the branched linker format simply because we wanted to clearly demonstrate effectiveness of EVCit linkers in a challenging case (i.e., both long spacer and cleavage mechanism were needed). Chemical synthesis of EVCit linkers is quite simple and does not require any unique chemical modification to the antibody or payload moiety. Given this point, this “just one more natural amino acid” strategy can most likely be adapted with ease for most ADCs that have been constructed using conventional linkers. This simple modification will allow researchers to design a variety of novel cleavable ADCs without worrying about premature cleavage in mouse models, which has been a huge obstacle in designing VCit-based ADCs. In particular, the EVCit linker will exhibit its full potential in conjugation at highly exposed sites where conventional VCit linkers are readily cleaved in mouse circulation (e.g., sortase-mediated conjugation and aldehyde tag-based conjugation at the C-terminus of the antibody heavy chain). Furthermore, as the other reviewers pointed out, the high polarity of EVCit is another advantage over conventional VCit because hydrophilicity is a key to prevent protein aggregation and fast clearance. Thus, our technology will be still beneficial when relatively protected conjugation sites are used for ADC construction. We are currently testing a variety of EVCit ADCs in combination with other conjugation chemistries and other classes of payload. These efforts will be published in the near future.

Comment 4: There are minor typos on pg 3, line 25, pg 4, line 48 and pg 8, line 137.

Reply 4: We apologize for the oversight. We have made necessary corrections or performed a grammar check.

[End of response to Reviewer 2]

<Reviewer 3>

Overall Comment: Tsuchikama and coworkers have written a very compelling manuscript describing a new variation on a classic ADC protease-cleavable linker that is stable to carboxylesterases. This improved stability results in improved pharmacokinetics as well as increased therapeutic efficacy. While this work was inspired by a publication from Dorywalska and coworkers at Pfizer to improve peptide linker stability, that work coupled improvements in peptide stability to site of conjugation and was limited in scope and applicability. Conversely, the impact of the current manuscript is high as it has

broad utility. The EVCit linker is stable even in linker types that are likely highly solvent exposed, making it generally useful for a variety of conjugation, spacer, and payload types. This makes the work of high interest to the ADC field and also of interest to an even broader field of targeted delivery (both therapeutics and imaging agents). I think it is virtually guaranteed that this paper will influence the field and that EVCit will be the linker selected for many future test ADCs. An additional advantage of this linker is the increased hydrophilicity accompanying the incorporation of glutamic acid. This is supported by HIC data and hopefully in future work by examples of rescuing linker-drugs that fail due to aggregation as they have done here rescuing ADCs that fail due to poor PK caused by instability.

The paper was extremely well written and referenced. Statistical analysis was provided for all appropriate data including half-lives, cytotoxicity, stability (in vitro and in vivo), and efficacy.

Reply to Overall Comment: We thank the reviewer for taking the time to review our manuscript. We appreciate this expert reviewer's detailed summary of this manuscript and the reviewer's enthusiasm regarding the novelty of our work.

Comment 1: The one suggestion I have is regarding the authors focus on cathepsin B as the protease primarily responsible for cleavage of the peptide linker. Recent work (Ref. 40) indicates that other cysteine cathepsins are also responsible for this proteolytic activity. The authors indicate that cleavage by CatB of the different linkers is equivalent (Supp. Fig. 5). While this claim could be solidified by testing cleavage with more than one (or a mixture of cathepsins) or even a lysosomal fraction (commercial preps are available), their in vitro cleavage result is appropriately supported by ADC activity data. The authors should at least address the complexity of the issue of proteolytic cleavage in the text in that multiple proteases might ultimately be responsible.

Reply 1: We appreciate the reviewer's astute comment and suggestion on the complexity of cathepsin-mediated ADC linker cleavage. We agree with this comment and we have changed some of "cathepsin B" in the manuscript to "cathepsin(s)". According to this suggestion, we performed additional cleavage assays using cathepsins L and S (see newly added Supplementary Fig. 5b). As reported in Ref. 40, these cathepsins can compensate for the loss of cathepsin B in tumor cells and indeed can cleave the valine-citrulline sequence. Interestingly, VCit ADC **3a** was slightly more sensitive to cathepsin L-mediated cleavage than EVCit **3c**. Cathepsin S cleaved both linker systems at similar rates. In addition, both sequences were cleaved at similar rates in a mixture of cathepsins B, L, and S. Thus, while highly responsive to cathepsin B-mediated cleavage, EVCit was not necessarily more sensitive to cleavage by other cathepsins than VCit. As reported in the manuscript, no significant enhancement of *in vitro* cell killing potency was observed in EVCit ADC **3c**. Considering this point along with the observed similar sensitivity to mixed cathepsins, there may be no distinct difference in the rate of lysosomal cleavage between the VCit and EVCit sequences. To incorporate this new finding, we have added the description below to the Results section:

"The ADCs were also tested for responsiveness to cathepsins L and S, which are also responsible for lysosomal cleavage of VCit linkers³⁴ (Supplementary Fig. 5b). Interestingly, VCit ADC **3a** was slightly more sensitive to cathepsin L-mediated cleavage than EVCit **3c**. Cathepsin S cleaved both linker systems at similar rates. In addition, both sequences were cleaved at almost equivalent rates in a mixture

of cathepsins B, L, and S. Thus, while highly responsive to cathepsin B-mediated cleavage, EVCit was not necessarily more sensitive than VCit to cleavage by other cathepsins.”

“Given that multiple proteases including cathepsins are most likely responsible for lysosomal processing of VCit-type linkers³⁴, there might be no distinct difference in the rate of lysosomal cleavage between the VCit and EVCit sequences. This speculation is also supported by our observation that both VCit and EVCit linkers were processed at almost equal rates in the presence of multiple cathepsins (Supplementary Fig 5b).”

Ref. 34 cited here is Ref. 40 in the original manuscript. We have updated the reference numbers affected by these additions.

Note: the relatively low reactivity of cathepsin L may be derived from lot-to-lot variation among commercial products and could be enhanced by increasing the amount. However, we do not think this issue is a critical factor for the difference in cleavage rate between both ADCs.

[End of response to Reviewer 3]

REVIEWERS' COMMENTS:

Reviewer #1 (Remarks to the Author):

The authors have adequately addressed my concerns.

Reviewer #2 (Remarks to the Author):

The authors have made the appropriate edits and changes to the manuscript. It can be accepted for publication.

Reviewer #3 (Remarks to the Author):

I think the authors did an excellent job addressing the reviewers concerns and the additional experiments performed and data and discussion added improved clarity as well as helping further establish novelty and superiority over prior work. I think the paper is acceptable for publication as is.